# Transfer of View-manifold Learning to Similarity Perception of Novel Objects

**Xingyu Lin, Hao Wang**
Department of Computer Science
Peking University
Beijing, 100871, China
`{sean.linxingyu, hao.wang}@pku.edu.cn`

**Zhihao Li, Yimeng Zhang**
Department of Computer Science
Carnegie Mellon University
Pittsburgh, PA 15213, USA
`{zhihaol, yimengzh}@andrew.cmu.edu`

**Alan Yuille**
Department of Cognitive Science
John Hopkins University
Baltimore, MD 21218, USA
`alan.yuille@jhu.edu`

**Tai Sing Lee**
Department of Computer Science
Carnegie Mellon University
Pittsburgh, PA 15213, USA
`tai@cnbc.cmu.edu`

## ABSTRACT

We develop a model of perceptual similarity judgment based on re-training a deep convolution neural network (DCNN) that learns to associate different views of each 3D object to capture the notion of object persistence and continuity in our visual experience. The re-training process effectively performs distance metric learning under the object persistency constraints, to modify the view-manifold of object representations. It reduces the effective distance between the representations of different views of the same object without compromising the distance between those of the views of different objects, resulting in the untangling of the view-manifolds between individual objects within the same category and across categories. This untangling enables the model to discriminate and recognize objects within the same category, independent of viewpoints. We found that this ability is not limited to the trained objects, but transfers to novel objects in both trained and untrained categories, as well as to a variety of completely novel artificial synthetic objects. This transfer in learning suggests the modification of distance metrics in view-manifolds is more general and abstract, likely at the levels of parts, and independent of the specific objects or categories experienced during training. Interestingly, the resulting transformation of feature representation in the deep networks is found to significantly better match human perceptual similarity judgment than AlexNet, suggesting that object persistence potentially could be a important constraint in the development of perceptual similarity judgment in our brains.

## 1 INTRODUCTION

The power of the human mind in inference and generalization rests on our brain's ability to develop models of abstract knowledge of the natural world (Tenenbaum et al., 2011). When shown novel objects, both children and adults can rapidly generalize from just a few examples to classify and group them based on their perceptual similarity. Understanding the processes that give rise to perceptual similarity will provide insight into the development of abstract models in our brain. In this paper, we explored computational models for understanding the neural basis of human perceptual similarity judgment.

Recent deep convolutional neural networks (DCNNs) have produced feature representations in the hidden layers that can match well with neural representations observed in the primate and human visual cortex. It was found that there is a strong correspondence between neural activities (neuronal spikes or fMRI signals) and the activities of the deep layers of deep networks (Agrawal et al., 2014; Khaligh-Razavi & Kriegeskorte, 2014; Yamins et al., 2014), suggesting that deep neural networks have in fact learned meaningful representations that are close to humans', even though the neural

networks are trained for object classification in computer vision. Cognitive neuroscientists have started to explore how the representations learned by deep networks can be used to model various aspects of human perception such as memorability of objects in images (Dubey et al., 2015), object typicality (Lake et al., 2015), and similarity judgment (Peterson et al., 2016; Kubilius et al., 2016). Certain correspondence between deep net representations and human experimental results are found. In particular, Peterson et al. (2016) found that human similarity judgment on a set of natural images might be similar to the feature representations in deep networks after some transformation.

The DCNNs that neuroscientists and cognitive scientists have studied so far, such as AlexNet (Krizhevsky et al., 2012), were trained with static images with the goal of classifying objects in static images into different categories. Perceptual similarity judgment is obviously closely related to the mechanisms used in object classification—we classify objects with similar attributes and appearances into the same class, and thus object classification rests in part on our perceptual similarity judgment and relies on physical, semantic abstract attributes common to objects in each class. Our perceptual similarity judgment might also be tied to our need for individual object recognition—after all, we might want to recognize an individual person or object, not just a class. It is obviously important to be able to recognize one's own child or the cup one is using. The need to recognize an individual object, independent of view points, requires fine discrimination of details, and might also be a very potent force for shaping our perceptual similarity judgment's machinery.

The development of invariant object recognition has often been attributed to object continuity or persistence in our visual experience. When we see an object, we tend to see it from different angles over time, as we walk by or around it, or directly manipulate it. This temporal persistence of objects allows our visual system to associate one view of an object with another view of the same object experienced in temporal proximity, as were proposed in slow-feature analysis (Wiskott & Sejnowski, 2002) or memory trace models (Perry et al., 2006) in computational neuroscience for learning translation and rotation invariance in object recognition. Object persistence as a term in psychology sometimes refers to people's knowledge or belief on the continual existence of an object even when it is occluded and invisible from view. Here, we use it to more generally to denote the temporal persistence of an object in our visual experience. We propose to incorporate the object continuity or persistence constraint in the training of DCNN, and investigate what new abstraction and capability such a network would develop as a consequence. We also evaluate the behaviors of the resulting network to see if they match the data on human perceptual similarity judgment of novel objects in an earlier study (Tenenbaum et al., 2011).

We retrain a DCNN with object persistence constraints, using rendered 3D objects. We call this retrained network Object Persistence Net (OPnet). During training, we utilize a Siamese network architecture for incorporating object persistence constraints into the network. We demonstrated that multi-view association training with a relatively small set of objects directly affects similarity judgment across many classes of objects, including novel objects that the network has not seen before. Our contribution is to demonstrate the surprising transfer of learning of similarity judgment to untrained classes of objects and a variety of completely artificial novel objects. We analyzed the view-manifold fine-tuned with object persistence constraints to understand what changes have taken place in the feature representation of the OPnet that has resulted in the development of this remarkable transfer of perceptual similarity judgment to novel objects.

Creating large sets of human labeled data on object similarity judgement is expensive. There has been a recent trend in exploring inherent information as supervisory signal, including using cycle consistency for learning dense correspondance(Zhou et al., 2015), camera motion for foreground segmentation(Zeng et al., 2016) and context information(Doersch et al., 2015). Among these, most related to our study is the work of Wang & Gupta (2015) utilizing visual tracking results as supervisory signals, which is an object persistence or continuity assumption, to learn deep networks without explicit object labels. While the tracked patches can be partly regarded as multi-view images, the changes in views tend to be very limited. In comparison, we used graphics rendered multi-view images as object persistency constraint. Such a clean setup is necessary for us to study the effect of object persistency constraint on novel objects, as well as the transferability of view-manifold learning to similarity perception.

Recent approaches in representation learning of 3D shapes are also related to our work. Generative models such as (Wu et al., 2016) and (Tatarchenko et al., 2015) learn a vector representation for generation of 3D shapes. Other approaches learn an embedding space for multi-view object retrieval

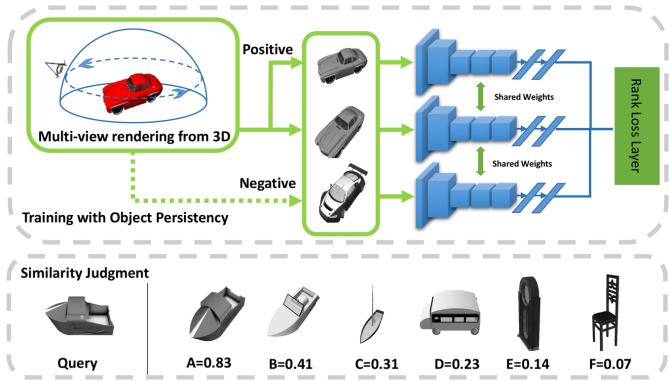

Figure 1: Framework for training and testing the network utilizing object persistence. For training (upper panel), we first render multiple views for each object and arrange them into triplets containing a similar pair and a dissimilar pair as input to a Siamese network architecture. For testing (lower panel), when given a query image, the network computes a similarity score for each of the candidate images. The lower panel shows some example similarity scores given by our OPnet, where different views of the same object are considered the most similar, followed by different objects in the same category, and finally those objects belonging to different categories of least similarities with the query image.

(Guo et al., 2016) or for cross-view image and shape retrieval(Li et al., 2015). While these works explored training with multi-view images, they did not constrain the view points in a continuous way and most importantly, the transferability to judgement of novel objects of novel classes were not studied. We evaluate the performance of the approach with Li et al. (2015) in our tasks for comparison. That approach learned an embedding space of 3D shapes and used CNN for image embedding for the purpose of image purification.

## 2 APPROACH AND METHODS

We take a standard CNN (AlexNet), that has already learned good feature representations for object classification, and retrain the network in a Siamese triplet architecture with object persistence constraints using multi-view images rendered from a set of 3D object models in ShapeNet.

### 2.1 OBJECT PERSISTENT NET (OPNET)

To study the impact of object persistence constraint in the development of perceptual similarity judgment, OPnet utilizes a Siamese triplet architecture. This triplet architecture can be visualized as three baseline CNN towers that share the same parameters (Figure 1). In implementation, it is just one single CNN applied to three images, two of which are considered more "similar" than the third "different" one. Conceptually, our OPnet tries to bring the feature representations of the two "similar" images together, and drive apart the representation corresponding to third "different" image. The architecture and the initial weights of the baseline CNN is same as those of of AlexNet trained on ImageNet (Deng et al., 2009). To train our OPnet with triplet input $(X_i, X_i^+, X_i^-)$, we present two views of the same 3D object to two base networks as $(X_i, X_i^+)$, and a view of a different object to the third base network as $X_i^-$. Object persistence means that given $(X_i, X_i^+, X_i^-)$, we try to push the representations for views of the same object $(X_i, X_i^+)$ to be close and make them away from the representation for the different object $X_i^-$. We minimize the loss function with a hinge loss term:

$$\min_W \frac{\lambda}{2} \parallel W \parallel_2^2 + \sum_{i=1}^{N} \max\{0, D(X_i, X_i^+) - D(X_i, X_i^-) + M\},$$

$$D(X_1, X_2) = 1 - \frac{f(X_1) \cdot f(X_2)}{\parallel f(X_1) \parallel \cdot \parallel f(X_2) \parallel},$$

where $\lambda$ is the weight decay and $W$ denotes the weights of the network. $f(\cdot)$ is the CNN representation output as a function of an input image, and $M$ denotes the margin parameter. The margin is a threshold

to decide whether the two views are considered similar or not. The higher the margin, the more we are forcing the network to develop a uniform representation for multiple views of the same object, relative to views of another object. $D$ is the cosine distance function for a pair of features.

The different objects in principle could be from the same category or from different categories. During training, we constrain the "different object" to be another 3D object from the same category to push apart more forcefully the feature representations of objects from the same category, resulting in view-invariant object discrimination within the same category. We expect the result of this training to create a view-manifold for each individual object—views within the same manifold are considered to be "similar" and closer together because they belong to the same object.

## 2.2 DISTANCE METRIC LEARNING

Our Siamese triplet approach transforms the view-manifold of the original baseline network, so that different views of the same object are considered similar and become closer in the feature representation space. Thus, it can be viewed as a form of distance metric learning (DML), which is a set of methods that learn a transformation from the input space to a feature space. The Siamese network has been a popular distance metric learning method, used in signature verification (Bromley et al., 1993), learning invariant mapping (Hadsell et al., 2006), face verification (Chopra et al., 2005), unsupervised learning (Wang & Gupta, 2015) or image similarity ranking (Wang et al., 2014). In these works, the definition of similarity for DML comes from the semantic labeling like class label. In our work, the similarity is defined by the object persistence constraints, obtained during the rendering of 3D models and providing a continuous trajectory for each single object. Besides, the large variation of the 2D appearance induced by 3D rotation prevents our network from learning trivial global templates, but induces it to learn features that are more generalized and thus transferable more easily to novel objects.

DCNNs, such as AlexNet, pre-trained on large dataset, have developed useful feature representations that can be fine-tuned for other specific tasks (Donahue et al., 2014; Qian et al., 2015; Karpathy et al., 2014). However, the pre-training of DCNN involves class labels as teaching signals. During pretraining, the network learns to throw away much information to extract invariants for classification. On the other hand, DML approaches are able to develop feature representations that preserve more fine-grained features, as well as intra- and inter-class variations.

## 2.3 RENDERING MULTI-VIEW DATASETS FOR SIMILARITY JUDGEMENT TRAINING

To allow the network to learn features under the object persistence constraints and develop a similarity judgment that can transfer, we create one set of data for training and five sets of novel objects for testing of the transferability. To focus our study on the network's ability to perceive 3D spatial relations and features of individual objects, we grayscale our images during rendering to eliminate the impact of color. For the same reason, we do not add any backgrounds.

We render multi-view images of individual objects from 7K 3D CAD models of objects in ShapeNet (Chang et al., 2015). The 7K models belong to 55 categories, such as cars and chairs. For each model, we render 12 different views by rotating the cameras along the equator from a $30°$ elevation angle and taking photos of the object at 12 equally separated azimuthal angles (see Fig. 1). We use the rendering pipeline in Blender, an open source 3D graphics software, with a spotlight that is static relative to the camera.

For training, we sample 200 object models from 29 categories of ShapeNet. 20 of these object models from each category are saved for cross validation. For testing, we make the assumptions that (1) views of the same object are perceived to be more similar when compared to views of a different object, and (2) views of objects in the same category are perceived to be more similar than views of objects from different categories. These assumptions are consistent with findings in earlier studies on similarity judgment in human (Quiroga et al., 2005; Erdogan et al., 2014; Goldstone, 2013). Since we render images based on CAD models, we can control the variations to create a large dataset that can approximate ground-truth data for similarity judgment for our experiments without resorting to large-scale human judgment evaluation. All the objects in the following five test sets are novel objects in the sense that they are not used in training.

**Novel instance**: Created by rendering additional 20 novel objects from each of the 29 categories used in training the OPnet. This is used to test the transfer of view-manifold learning to novel objects of the same category. The task is not trivial due to the large intra-category variation existing in the ShapeNet.

**Novel category**: Created by rendering objects from 26 untrained categories. This is a more challenging test of the transfer of view-manifold learning to novel categories.

**Synthesized objects**: Created by rendering a set of 3D models we synthesized. These are textureless objects with completely novel shapes. The dataset consists of 5 categories, with 10 instances for each category. Within each category, the objects either have similar local parts, or have the same global configuration, based on human judgment. This is an even more challenging test, as these synthesized objects are not in the ImageNet or ShapeNet.

**Pokemon** Created by 3D models of Pokemon dataset. Pokemons are cartoon characters with certain evolution relationships with each other, which provides an alternative measurement of similarity. This test evaluates the transfer of learning to novel objects with different styles and more complicated textures. We collected 438 CAD models of Pokemon from an online database. We divide these models into 251 categories according to their evolution relationships, with most of these categories containing only 2 to 4 objects. Pokemons of the same category look more similar on average due to their "genetic linkage".

**Tenenbaum objects** This test set contains novel objects from Tenenbaum et al. (2011), where the ground truth is based on human similarity judgment.

## 2.4 SIMILARITY JUDGMENT EVALUATION

The similarity score between a query image and a candidate image is computed as 1 minus the cosine distance of the feature representations of the query and candidate pair, and higher score means higher similarity. Given a test set containing objects of multiple categories, we evaluate the OPnet via two retrieval tasks: object instance retrieval and categorical retrieval. In the object instance retrieval task, for each image $P$ containing object $O$ of category $C$ in the test set, the network is asked to rank all other images in $C$, such that images for $O$ should have higher similarity score than images for other objects in $C$. In the categorical retrieval task, for each image $P$ of category $C$, the network is asked to rank all other images, such that images in category $C$ should have higher score than images not in $C$. Here we are indirectly utilizing the human perception information, as categories are defined by human perception based on their similarity in shapes or functions.

## 2.5 IMPLEMENTATION DETAILS

We use Caffe (Jia et al., 2014) for training the networks. The base network of the OPnet is modified from the AlexNet architecture, where we drop the last fully connected layer (fc8) and replace the softmax loss with our triplet hinge loss. The network is initialized by weights pre-trained on ImageNet. The objective is optimized using mini-batch stochastic gradient descent (SGD) and we fine-tune the network for all layers. For each pair of positive example $(X, X^+)$, we select two hard negative examples $X^-$ which give the highest loss (similar in (Wang & Gupta, 2015)) and another two randomly from within the mini-batch. Starting with a learning rate of 0.01, we decrease it by a factor of 10 every 8K iterations and with a momentum of 0.9. We stop the training at 20K iterations. Weight decay is set to 0.0005. We set the margin parameter $M$ to 0.1 by cross validation.

## 3 EXPERIMENTAL RESULTS

We compare HoG feature representation (Dalal & Triggs, 2005) and four deep learning networks: 1) OPnet, 2) AlexNet pre-trained on ImageNet, 3) An AlexNet fine-tuned for classification on ShapeNet data, denoted as "AlexNetFT", 4) The joint embedding model by Li et al. (2015). In AlexNetFT, we replace the original fc8 layer with a fully connected layer with 29 output units and fine-tune the last two fully connected layers (fc7, fc8) with cross-entropy loss. The AlexNetFT model is trained with the same data we used for training the OPnet. The joint embedding model was pre-trained on 6700 shapes in the chair category of ShapeNet. For the first three deep models, we use the fc7 layer as the feature representation and cosine distance to compute distance between feature representations. We

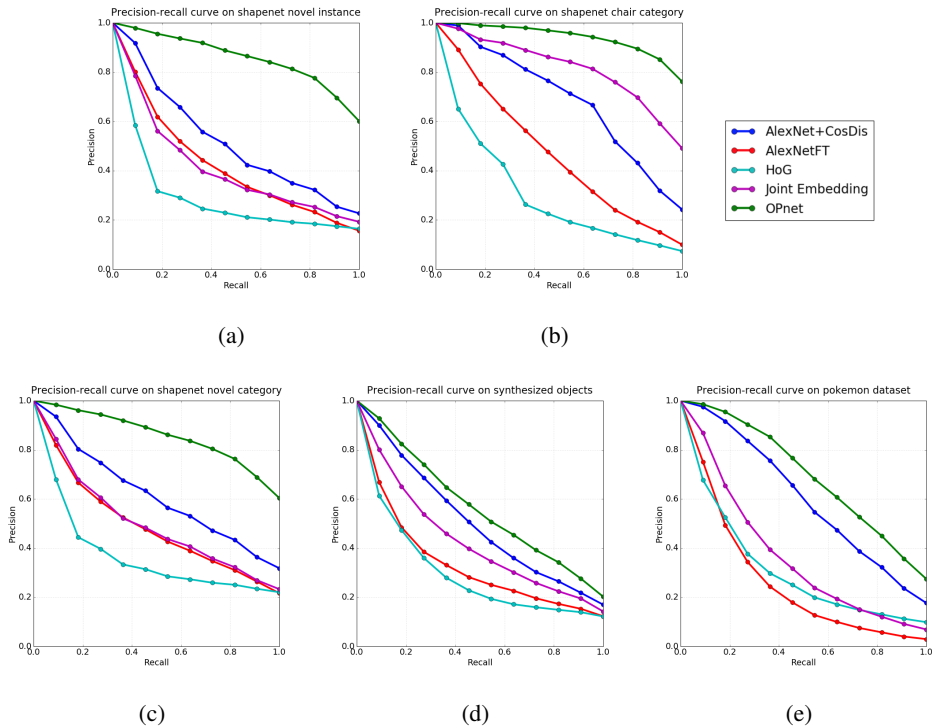

Figure 2: The precision-recall curves for the object instance retrieval task on different datasets.

|  | Novel instance | Novel category | Synthesized objects | Pokemon | Chair |
|---|---|---|---|---|---|
| HoG | 0.316 | 0.391 | 0.324 | 0.332 | 0.322 |
| AlexNetFT | 0.437 | 0.503 | 0.356 | 0.287 | 0.478 |
| AlexNet+CosDis | 0.529 | 0.623 | 0.517 | 0.607 | 0.686 |
| AlexNet+EucDis | 0.524 | 0.617 | 0.514 | 0.591 | 0.677 |
| OPnet | **0.856** | **0.855** | **0.574** | **0.697** | **0.938** |
| Joint-embedding | 0.429 | 0.513 | 0.443 | 0.387 | 0.814 |

Table 1: Mean Average Precision for the object instance retrieval task over all test sets.

also show results based on AlexNet feature representation both in terms of Eculidean distance and cosine distance measures, denoted as AlexNet+EcuDis and AlexNet+CosDis. Comparison of feature representations from different layers are shown in Appendix B. We show the results for the instance retrieval task in Figure 2 and Table 1. The precision measure reflects the accuracy of the model's similarity judgment, with the two assumptions given in section 2.3.

On similarity judgment of novel objects from both the trained and untrained categories, OPnet significantly outperforms AlexNet and AlexNetFT, with an increased Mean Average Precision of at least 23%. The improvement is due to OPnet's gains in ability in discriminating different objects inside one category regardless of their viewpoints, while recognizing different views of the objects to be similar. For novel shapes in artificial synthesized objects and Pokemons, OPnet still shows an increased MAP of at least 6% (or 15% decreased error rate for the Pokemon test). This shows that the similarity judgment resulting from view manifold learning is valid not only for the trained objects or just to the objects in the same data set, but generalizable to other classes of objects. This suggests the learned feature representations are more abstract and general, allowing the transfer of the learning to substantially different datasets and novel objects, to a degree that is not well known or well studied in computer vision.

We compare OPnet with the joint embedding approach on the chair category of ShapeNet, shown in Figure 2b. Both networks are trained with the chair category and are tested on novel chairs. OPnet outperforms the joint embedding approach by a large margin, showing that a better instance level

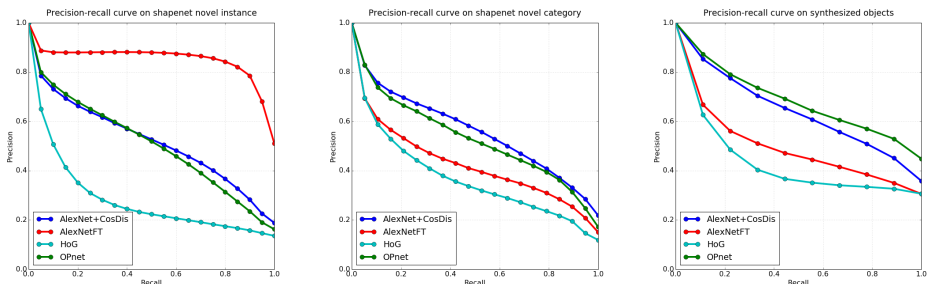

Figure 3: The precision-recall curves for the category level retrieval task. The three figures show the network's performance on the ShapeNet dataset with novel instance, novel category and synthesized objects respectively.

discrimination is achieved using object persistence training, compared to using known shapes as anchor points for image embedding. Furthermore, because the joint embedding approach would need to be trained for each specific category, it does not perform well on novel categories.

When we fine-tuned AlexNet for classification of the 29 trained categories, the resulting AlexNetFT's feature representation actually performs the worst, compared to OPnet and the original AlexNet, on the instance similarity judgment or retrieval tasks. When a network is trained to perform classification, it learns to ignore subtle differences among objects in the same category. The fewer categories a network is trained on, the more the instance level similarity judgment will be compromised. This loss of the generality of its feature representation compromises its transferability to novel objects in other classes.

We notice that the performance gain for the OPnet is most significant in the ShapeNet dataset and the gap becomes small for the synthesized and Pokemon dataset. This shows OPnet's certain overfitting to the bias in ShapeNet, as the synthesized object dataset contains textureless objects and Pokemon dataset contains mainly human-like characters that are not in ShapeNet.

Categorical retrieval provides another measure of the network's performance in similarity judgment. In this test, we randomly sample 20 categories each from the novel instance test and the novel category test, with 20 object instances drawn from each category. For the synthesized object test set, we test all 5 categories and each with 10 instances. For each instance, a single random view is provided. The results are shown in Figure 3. Despite the fact that AlexNet knows more about the semantic features of each category, our OPnet still achieves comparable results. OPnet here shows an improved ability in similarity judgment at the categorical level. On our artificially synthesized object dataset, where all three networks have no prior experience, OPnet performs better than AlexNet. AlexNetFT performs extremely well on trained categories likely because it is overfitted to the limited trained objects, even though it uses the same amount of data. This overfitting problem shows that training with only class labels might not preserve the essential information to develop transferable general feature and abstract feature representation, especially with limited training dataset.

## 3.1 CORRELATION WITH HUMAN PERCEPTION

Using the novel objects from Tenenbaum et al. (2011), we are able to compare our networks with human similarity perception. We collect 41 images from the paper, one image per object. A pairwise similarity matrix is calculated based on the cosine distance of their feature representations. We can then perform hierarchical agglomerative clustering to obtain a tree structure, using the Nearest Point Algorithm. That is, for all points $i$ in cluster $u$ and points $j$ in cluster $v$, the distance of the two clusters are calculated by $\mathrm{dist}(u,v) = \min(D(u[i], v[j]))$, where $D(\cdot)$ is the cosine distance function. We merge two clusters with the shortest distance successively to construct the tree. The tree based on human perception is constructed by giving human subjects all the images and asking them to merge two clusters that are most similar each time, similar to the hierarchical agglomerative clustering algorithm. Results are shown in Figure 4.

In order to quantitatively measure the similarity between the trees output by neural networks and the one based on human perception, we calculate the Cophenetic distances on the tree for each pair of

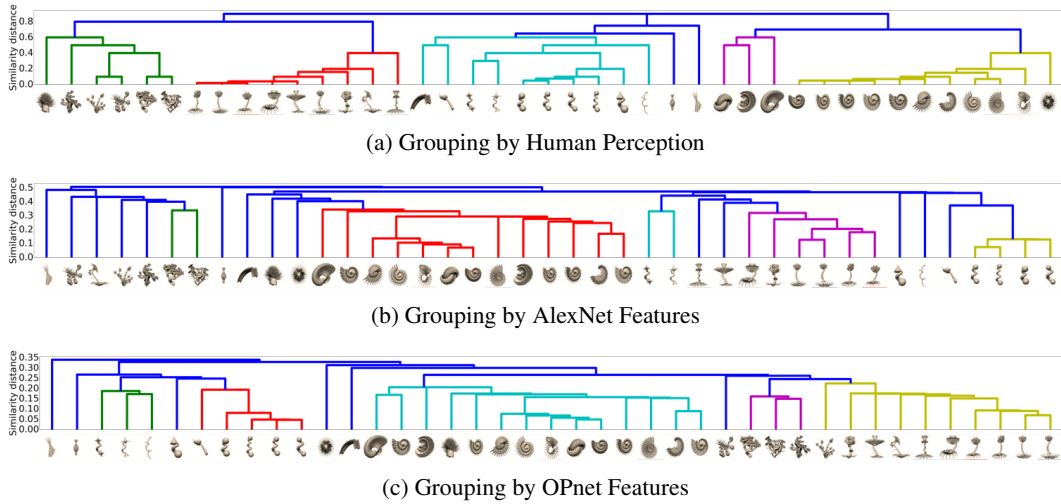

(a) Grouping by Human Perception

(b) Grouping by AlexNet Features

(c) Grouping by OPnet Features

Figure 4: Hierarchical clustering of the alien objects, based on (a) human perceptions, (b)A lexNet features and (c) OPnet features. The dendrograms illustrate how each cluster is composed by drawing a U-shaped link between a cluster and its children. The height of each U-link denotes the distance between its children clusters when they are merged.

object. For object $i$ and $j$, the Cophenetic distances $t_{i,j}$ are defined as $t_{i,j} = \text{dist}(u, v), i \in u, j \in v$, where $u,v$ are clusters connected by U-link. Finally, we can evaluate the similarity of the two trees by calculating the Spearman's rank correlation coefficient. In the experiment, the Spearman correlation is 0.460 between AlexNet and the human perception and 0.659 between OPnet and the human perception, meaning that our OPnet, trained with object persistence constraints on a relatively small set of objects, automatically yielded a higher match to the human perceptual similarity data. This finding provides some support to our conjecture that object persistence might play an important role in shaping human similarity judgment.

## 3.2 STRUCTURES OF THE VIEW MANIFOLD

We study the feature representations in these networks and their transformation induced by the object persistence constraints to understand how the changes in similarity judgment performance come about. As our network uses cosine distance in the feature space as similarity measure, we study how this measure changes in the view-manifold of the same object and between views of different objects.

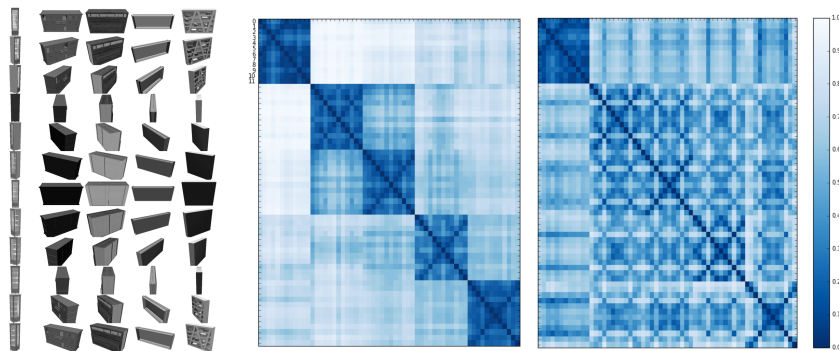

Figure 5: Distance measures for 5 cabinet objects. Lighter pixels mean larger distance. On the left is the objects each with 12 views, whose similarity distance between each other we are interested in. In the middle and the right is the cosine distance of the ouput features of OPnet and AlexNet respectively. The element on the $i^{th}$ row and the $j^{th}$ column stands for the cosine distance between the $i^{th}$ and $j^{th}$ image. The $i^{th}$ image is rendered from $[i/12]^{th}$ object and $(i \bmod 12)^{th}$ view.

We first visualize the pairwise similarity distance matrix of AlexNet and OPnet in Figure 5. We randomly choose 5 objects from the cabinet category for illustration. Each object has 12 views that the network has never seen before. Images are arranged first by different object instances (in columns) then by views (in rows). Many properties of the view manifolds are revealed. First, for the matrix of OPnet, we can see clearly five dark blocks formed in the diagonal, each standing for the strong similarity (small distance) among the different views of the same cabinet. The dark block means that OPnet is associating different views of the same object together, reducing intra-object distance relative to inter-object distance. In this way, the similarity judgment of the OPnet becomes more viewpoint independent. On the other hand, the similarity matrix of AlexNet shows a variety of patterns across all objects within the same category. A closer look at these patterns suggests that AlexNet first forms groups by certain views (e.g. side-views), and then by objects, resulting in a more viewpoint dependent similarity measure that is poor in discriminating objects within a category. Second, even though OPnet groups different views together, the view-manifold has not degenerated into a single point. Certain patterns can be seen inside each dark block of OPnet's matrix, forming a hierarchical structure: different views of the same object are more similar to each other than to another object and some rotations in angle are considered more similar than others. To illustrate how the view manifolds have contracted but not completely degenerated, we randomly sample objects from the novel instance test set and use TSNE (Maaten & Hinton, 2008) to plot them in 2D, as shown in Figure 6. We can see clearly that different views of the same object are considered more similar in the feature space, and objects form tight and distinct clusters. We borrow a measurement from Linear

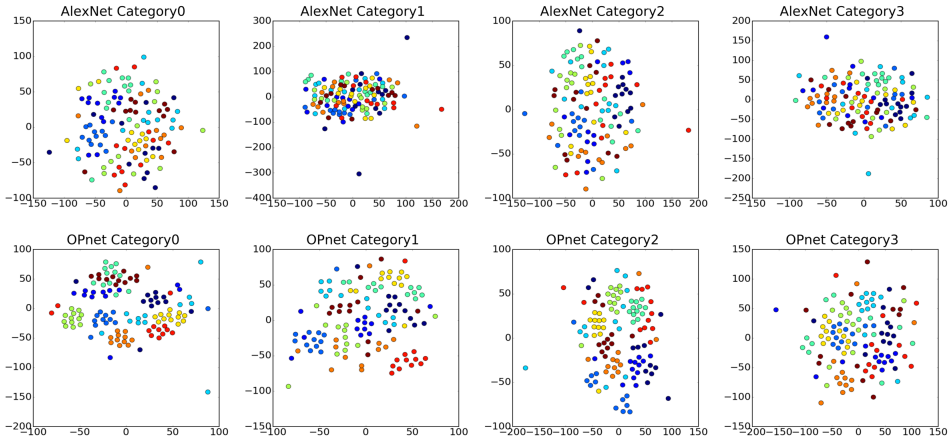

Figure 6: TSNE visualization of the features produced by AlexNet and OPnet, on four categories. Each point represents a view of an object. Different colors represent different objects.

Discriminant Analysis (LDA) to evaluate how tightly different views of the same object are clustered together, relative to the distance among different objects within the same category. Formally, let $S$ be the set of all the objects inside one category and $c$ be the set of all views for one object, $\bar{x}$ be the center of all image features, and $\mu_c$ be the center for the object $c$. We then calculate the score for category $i$ using the following equation:

$$Score_i = \frac{\sigma_{inter\_instance}}{\sigma_{intra\_instance}} = \frac{\frac{1}{|S_i|}\sum_c ||\mu_{\mathbf{c}} - \bar{\mathbf{x}}||^2}{\frac{1}{|S_i|}\sum_{c \in S_i}\frac{1}{|c|}\sum_{\mathbf{x} \in c}||\mathbf{x} - \mu_{\mathbf{c}}||^2}$$

We then average over all the categories to get a score for each network. The higher the score is, the larger the inter-object distance is compared to intra object distance and the more closely different views of the same object are grouped together. In the experiment with the novel instance test set, OPnet's score is 0.535 whereas AlexNet's is 0.328, showing the different views of the same object are more similar than that between different objects due to the object persistence constraint.

## 4 CONCLUSION AND DISCUSSION

In this work, we fine-tune AlexNet with object persistence constraints in the framework of distance metric learning with a Siamese triplet. This fine-tuning modifies the view-manifold of the object

representation, bringing closer together the representations of an object in different views, driving apart representations of different objects in the same category, resulting in better intra-categorical object recognition, without compromising inter-categorical discrimination. We investigated whether this view-manifold learning results in an improvement in the network's ability to recognize the similarity of novel objects that have never been seen before by performing instance and categorical image retrieval on artificial novel objects or novel object classes, including a set tested in human similarity judgment. Interestingly, we find that AlexNet, with its rich feature representations, already perform similarity judgement significantly above chance, in the sense that different views of the same object are considered more similar to the views of another object in the same category, or objects in the same category are considered to be more similar than objects in different categories. Fine-tuning with the object persistence constraint significantly improves this "similarity judgement" among a variety of novel objects, suggesting the view manifold learning in the OPnet is accompanied by feature embeddings with more general and abstract attributes that are transferable, likely at the level of local object parts.

From a technical point of view, our OPnet performs better than earlier approaches (Li et al., 2015) in instance and categorical retrieval of novel objects. We have tested our approach with real image database (Geusebroek et al., 2005) and found it only yields a slight improvement over AlexNet. That database contains 1000 objects with different views but without categorical labels. OPnet's superiority over AlexNet lies in its better discrimination of objects within the same category. When objects are not organized in categories, i.e. when each object is essentially treated as a category, OPnet loses its advantages. In addition, there are more complex variations such as lighting and scale in real scene environments that our current OPnet has not considered. We plan to develop this model to discount additional nuisance variables and to develop or find database to explore the transferability of its view-manifold learning in more general settings.

Our work was motivated by our hypothesis that object persistence/continuity constraint in our visual experience might play a role in the development of neural representations that shape our similarity judgement of objects that we have not seen before. The fact that fine-tuning AlexNet with this additional constraint automatically yields a new view-manifold that match human similarity judgment data better than AlexNet lends some support to our hypothesis. However, more extensive testing with human perception ground-truth will be needed to fully confirm our hypothesis.

ACKNOWLEDGMENTS

Xingyu Lin and Hao Wang were supported by the PKU-CMU summer internship program. This work is supported by the Intelligence Advanced Research Projects Activity (IARPA) via Department of Interior/ Interior Business Center (DoI/IBC) contract number D16PC00007. The U.S. Government is authorized to reproduce and distribute reprints for Governmental purposes notwithstanding any copyright annotation thereon. Disclaimer: The views and conclusions contained herein are those of the authors and should not be interpreted as necessarily representing the official policies or endorsements, either expressed or implied, of IARPA, DoI/IBC, or the U.S. Government.

We thank Kalina Ko for helping us to construct part of the synthesized object database.

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

## APPENDIX A EXAMPLES OF SOME TOP RANKING RESULTS

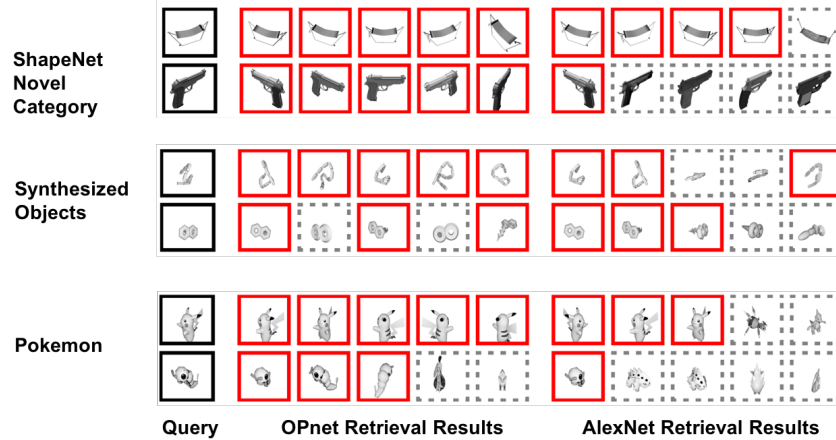

Figure 7: Examples of top instance retrieval results for AlexNet and OPnet. Images that are different views of the same object(which are considered more similar) are marked with red solid rectangle while views of other objects are marked with gray dashed rectangle. Obviously from the gun example we can see how the retrieval results for AlexNet are highly view-dependent.

## APPENDIX B INSTANCE RETRIEVAL RESULTS USING FEATURES FROM DIFFERENT LAYERS

As shown in many literatures (Massa et al., 2015; Aubry & Russell, 2015), features from different layers sometimes perform differently for a given task. For the instance retrieval task on the novel instance dataset of the ShapeNet, we compare OPnet and AlexNet using features from different layers, as shown in Figure 8. The accuracy of AlexNet is pretty flat up to conv3, and then keeps increasing until layer fc8 where the feature becomes categorical probability and not appropriate for instance level discrimination. On the other hand, the object persistence training gives a significant increase in accuracy in fully connected layers.

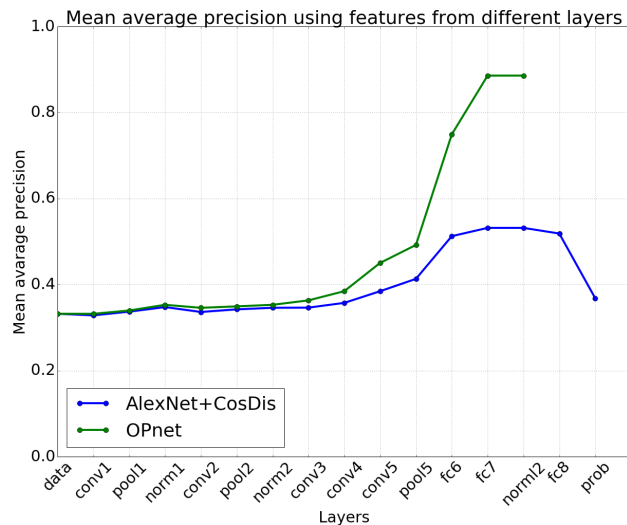

Figure 8: Instance Retrieval Results Using Features From Different Layers

