# Peer review of "Transfer of View-manifold Learning to Similarity Perception of Novel Objects"

_ICLR 2017 — accepted_

[Official Review · AnonReviewer3 · rating 7 · confidence 3 · 16 Dec 2016]
**interesting connections to human perception**
clarity 5 · meaningful comparison 4

On one hand this paper is fairly standard in that it uses deep metric learning with a Siamese architecture. On the other, the connections to human perception involving persistence is quite interesting. I'm not an expert in human vision, but the comparison in general and the induced hierarchical groupings in particular seem like something that should interest people in this community. The experimental suite is ok but I was disappointed that it is 100% synthetic. The authors could have used a minimally viable real dataset such as ALOI

[Official Review · AnonReviewer2 · rating 6 · confidence 4 · 17 Dec 2016]
**Nice form of supervision to explore**
clarity 5

I think learning a deep feature representation that is supervised to group dissimilar views of the same object is interesting. The paper isn't technically especially novel but that doesn't bother me at all. It does a good job exploring a new form of supervision with a new dataset. I'm also not bothered that the dataset is synthetic, but it would be good to have more experiments with real data, as well. 

I think the paper goes too far in linking itself to human vision. I would prefer the intro not have as much cognitive science or neuroscience. The second to fourth paragraphs of the intro in particular feels like it over-stating the contribution of this paper as somehow revealing some truth about human vision. Really, the narrative is much simpler -- "we often want deep feature representations that are viewpoint invariant. We supervise a deep network accordingly. Humans also have some capability to be viewpoint invariant which has been widely studied [citations]". I am skeptical of any claimed connections bigger than that.

I think 3.1 should not be based on tree-to-tree distance comparisons but instead based on the entire matrix of instance-to-instance similarity assessments. Why do the lossy conversion to trees first? I don't think "Remarkably" is justified in the statement "Remarkably, we found that OPnets similarity judgement matches a set of data on human similarity judgement, significantly better than AlexNet"

I'm not an expert on human vision, but from browsing online and from what I've learned before it seems that "object persistence" frequently relates to the concept of occlusion. Occlusion is never mentioned in this paper. I feel like the use of human vision terms might be misleading or overclaiming.

[Official Review · AnonReviewer1 · rating 5 · confidence 5 · 18 Dec 2016]
**Potentially interesting idea, but important references and baseline comparisons are missing.**

This paper proposes a model to learn across different views of objects.  The key insight is to use a triplet loss that encourages two different views of the same object to be closer than an image of a different object.  The approach is evaluated on object instance and category retrieval and compared against baseline CNNs (untrained AlexNet and AlexNet fine-tuned for category classification) using fc7 features with cosine distance.  Furthermore, a comparison against human perception on the "Tenenbaum objects” is shown.

Positives: Leveraging a triplet loss for this problem may have some novelty (although it may be somewhat limited given some concurrent work; see below).  The paper is reasonably written.

Negatives: The paper is missing relevant references of related work in this space and should compare against an existing approach.

More details:

The “image purification” paper is very related to this work:

[A] Joint Embeddings of Shapes and Images via CNN Image Purification. Hao Su*, Yangyan Li*, Charles Qi, Noa Fish, Daniel Cohen-Or, Leonidas Guibas. SIGGRAPH Asia 2015.

There they learn to map CNN features to (hand-designed) light field descriptors of 3D shapes for view-invariant object retrieval.  If possible, it would be good to compare directly against this approach (e.g., the cross-view retrieval experiment in Table 1 of [A]).  It appears that code and data is available online (

[Author Response · Xingyu Lin · 16 Jan 2017 (modified: 17 Jan 2017)]
**General Responses to the Reviewers:**

We thank all the official and unofficial reviewers for their extremely useful suggestions and are encouraged by the positive feedbacks. 

Following these suggestions, we have made a number of revisions and uploaded a new version of our manuscript, including adding important references, baselines and an experiment showing performance of features from different layers in Appendix B. We have colored the modifications red in order to provide an easier way to track the changes from the original submission.

[Final Decision · Program Chairs · 06 Feb 2017]
**ICLR committee final decision**

The paper proposes a model for multi-view learning that uses a triplet loss to encourage different views of the same object to be closer together then the views of two different objects. The technical novelty of the model is somewhat limited, and the reviewers are concerned that experimental evaluations are done exclusively on synthetic data. The connections to human perception appear interesting. Earlier issues with missing references to prior work and comparisons with baseline models appear to have been substantially addressed in revisions of the paper. We strongly encourage the authors to further revise their paper to address the remaining outstanding issues mentioned above.